# Loss of spatacsin impairs cholesterol trafficking and calcium homeostasis

Maxime Boutry[1,2,3,4,5,7], Alexandre Pierga[1,2,3,4], Raphaël Matusiak[1,2,3,4], Julien Branchu[1,2,3,4], Marc Houllegatte[1,2,3,4,5], Yoan Ibrahim[1,2,3,4], Elise Balse[6], Khalid-Hamid El Hachimi[1,2,3,4,5], Alexis Brice[1,2,3,4], Giovanni Stevanin[1,2,3,4,5] & Frédéric Darios [1,2,3,4]*

Mutations in *SPG11*, leading to loss of spatacsin function, impair the formation of membrane tubules in lysosomes and cause lysosomal lipid accumulation. However, the full nature of lipids accumulating in lysosomes and the physiological consequences of such accumulation are unknown. Here we show that loss of spatacsin inhibits the formation of tubules on lysosomes and prevents the clearance of cholesterol from this subcellular compartment. Accumulation of cholesterol in lysosomes decreases cholesterol levels in the plasma membrane, enhancing the entry of extracellular calcium by store-operated calcium entry and increasing resting cytosolic calcium levels. Higher cytosolic calcium levels promote the nuclear translocation of the master regulator of lysosomes TFEB, preventing the formation of tubules and the clearance of cholesterol from lysosomes. Our work reveals a homeostatic balance between cholesterol trafficking and cytosolic calcium levels and shows that loss of spatacsin impairs this homeostatic equilibrium.

[1] Sorbonne Université, UPMC Univ Paris 06, UMR S 1127, F-75013 Paris, France. [2] Inserm, U1127, F-75013 Paris, France. [3] CNRS, UMR 7225, F-75013 Paris, France. [4] Institut du Cerveau et de la Moelle Epinière, ICM, F-75013 Paris, France. [5] Ecole Pratique des Hautes Etudes, PSL Research University, Laboratoire de Neurogénétique, F-75013 Paris, France. [6] Sorbonne Université, UPMC Univ Paris 06, UMR S 1166, F-75013 Paris, France. [7]Present address: Cell Biology Program, Hospital for Sick Children, Peter Gilgan Centre for Research and Learning, Toronto, ON, Canada. *email: frederic.darios@upmc.fr

  **1**

Mutations in the *SPG11* gene are responsible for a severe form of hereditary spastic paraplegia characterized by bilateral weakness, spasticity in the lower limbs, as well as ataxia or cognitive impairment[1,2]. Most mutations are truncating mutations, suggesting that the symptoms are due to loss of function of the *SPG11* product, spatacsin[3]. Accordingly, knockout of *Spg11* in the mouse reproduces the main motor and cognitive symptoms observed in patients[4]. Studies in SPG11 patient fibroblasts and in *Spg11* knockout mice suggested that loss of spatacsin led to impaired function of lysosomes[4–6]. Lysosomes are organelles containing hydrolytic enzymes that notably fuse with endosomes or autophagosomes to allow degradation of their content. After the degradation step, new lysosomes can be reformed from the hybrid organelles[7,8]. Recycling of the lysosomal membrane after the termination of autophagy, known as autophagic lysosome recovery (ALR), relies on the formation of tubules on the lysosomes. This mechanism involves proteins that participate in membrane trafficking, such as clathrin and dynamin[9,10], but it also relies on spatacsin[11].

Analysis of *Spg11* knockout mice showed that the loss of spatacsin function led to progressive accumulation of lipids in lysosomes, both in neuronal and non-neuronal cells[4]. In particular, it was shown that loss of spatacsin led to lysosomal accumulation of glycosphingolipids in neuronal models[12]. Most lipids such as triacylglycerols, phospholipids, and gangliosides are degraded by the lysosomal hydrolases into their basic building blocks. The latter are then exported in the cytosol to be further degraded to fuel energy metabolism or can re-enter biosynthetic pathways[13]. In contrast, cholesterol is not degraded in the endolysosomal pathway, but it is exported out of this subcellular compartment. It is redistributed to the membranes of other subcellular compartments, placing lysosomes at a crossroad of cholesterol metabolism[14]. However, the molecular mechanisms by which cholesterol leaves late endosomes/lysosomes and reaches other subcellular compartments have been only partially characterized[15]. Furthermore, alteration of cholesterol trafficking is associated with many pathological conditions[16]. It is therefore important to explore the downstream consequences for cellular physiology of impaired cholesterol trafficking. Cholesterol has long been known to influence cellular calcium homeostasis, but little is known about the molecular mechanisms coupling change in cholesterol concentration to alterations of calcium signaling[17].

Here, we show that the loss of spatacsin function and the associated inhibition of tubule formation in late endosomes/lysosomes leads to the accumulation of cholesterol in this compartment, due to its impaired export out of the organelle. This results in a decrease in the level of plasma membrane cholesterol that disturbs intracellular calcium homeostasis. We demonstrate that the resulting modification in cytosolic calcium levels contributes to the impairment of lysosome tubulation and accumulation of cholesterol in late endosomes/lysosomes and that this process is TFEB-dependent.

## Results

**Tubules on lysosomes contributes to cholesterol clearance**. We analyzed the localization of lysosomes in control and spatacsin-deficient (*Spg11*$^{-/-}$) fibroblasts by LAMP1 immunostaining. *Spg11*$^{-/-}$ cells showed perinuclear accumulation of LAMP1-positive vesicles (Fig. 1a, b), a phenotype that has been linked to the accumulation of cholesterol in late endosomes and lysosomes[18,19]. We thus tested whether cholesterol accumulates in the late endosomes/lysosomes of *Spg11*$^{-/-}$ fibroblasts by monitoring intracellular localization of cholesterol with filipin, which stains free cholesterol (Fig. 1c), or the fluorescent probe derived from perfringolysin-O, GFP-D4[20] (Supplementary

Fig. 1a). The mean fluorescence intensity of filipin staining of whole cells was the same in *Spg11*$^{+/+}$ and *Spg11*$^{-/-}$ fibroblasts (Fig. 1d), a result confirmed by the biochemical analysis of cellular cholesterol content (Supplementary Fig. 1b). However, the proportion of cholesterol colocalized with late endosomes/lysosomes was significantly higher in *Spg11*$^{-/-}$ than control fibroblasts when monitored with filipin or GFP-D4 (Fig. 1e, Supplementary Fig. 1c). Since mutations in SPG11 cause neurodegeneration[3], we evaluated the impact of loss of spatacsin function on cholesterol distribution in neuronal models. Biochemical quantification showed that the amount of total cholesterol was similar in *Spg11*$^{+/+}$ and *Spg11*$^{-/-}$ neurons (Fig. 1f). We monitored whether cholesterol accumulates in the late endosomes/lysosomes of *Spg11*$^{-/-}$ neurons with the GFP-D4 probe or filipin staining (Fig. 1g, Supplementary Fig. 1d). Consistent with data obtained in fibroblasts, the proportion of cholesterol colocalized with late endosomes/lysosomes was significantly higher in *Spg11*$^{-/-}$ than control neurons, suggesting that cholesterol distribution was impaired in neurons in the absence of spatacsin (Fig. 1h, Supplementary Fig. 1d). We previously showed that loss of spatacsin induced the accumulation of gangliosides in lysosomes in neuronal models[12]. We tested whether cholesterol accumulation could be a consequence of the accumulation of gangliosides, by preventing their synthesis using miglustat. Inhibition of ganglioside synthesis did not prevent the accumulation of cholesterol in late endosomes/lysosomes (Supplementary Fig. 1d), suggesting that cholesterol accumulation is not a consequence of the accumulation of gangliosides.

Since the distribution of cholesterol, but not the total amount, was altered in the absence of spatacsin, we hypothesized that the trafficking of cholesterol could be disturbed. We monitored the trafficking of fluorescently labeled cholesterol. Control and *Spg11*$^{-/-}$ fibroblasts were incubated with low density lipoprotein (LDL) loaded with fluorescent cholesterol for two hours and chased for 24 h. We quantified the colocalization of fluorescent cholesterol with LAMP1 at several time points. During the first four hours, the proportion of fluorescent cholesterol colocalized with LAMP1 increased, consistent with the internalization of LDL, and there was no difference in the internalization of cholesterol between *Spg11*$^{+/+}$ and *Spg11*$^{-/-}$ fibroblasts. At longer chase times, there was a progressive decrease in the colocalization of fluorescent cholesterol and LAMP1 in control cells, consistent with the egress of cholesterol from late endosomes/lysosomes[21]. In contrast, the proportion remained stable in *Spg11*$^{-/-}$ cells (Fig. 1i), suggesting that the efflux of cholesterol from late endosomes/lysosomes was altered in the absence of spatacsin.

Spatacsin participates in the initiation of tubule formation on lysosomes[11]. Accordingly, we observed that *Spg11*$^{-/-}$ fibroblasts contained fewer lysosomes with tubules than *Spg11*$^{+/+}$ fibroblasts under basal condition when they were transfected with a vector expressing LAMP1-mCherry and analyzed by live imaging (Supplementary Fig. 2a, b). We tested whether the formation of tubules contributed to cholesterol clearance from lysosomes using siRNA to downregulate the clathrin heavy chain (Fig. 2a), a protein essential for the initiation of tubule formation on late endosomes/lysosomes[9]. Downregulation of the clathrin heavy chain in wild-type fibroblasts significantly decreased the number of tubules emanating from lysosomes and increased the proportion of cholesterol colocalized with the LAMP1-positive compartment under basal condition (Fig. 2b, c). Pulse-chase experiments of LDL loaded with fluorescent cholesterol showed that the efflux of cholesterol from late endosomes/lysosomes decreased when clathrin heavy chain was downregulated (Supplementary Fig. 2c). The scission of lysosome tubules requires dynamin[10], a binding partner of spatacsin[12]. The

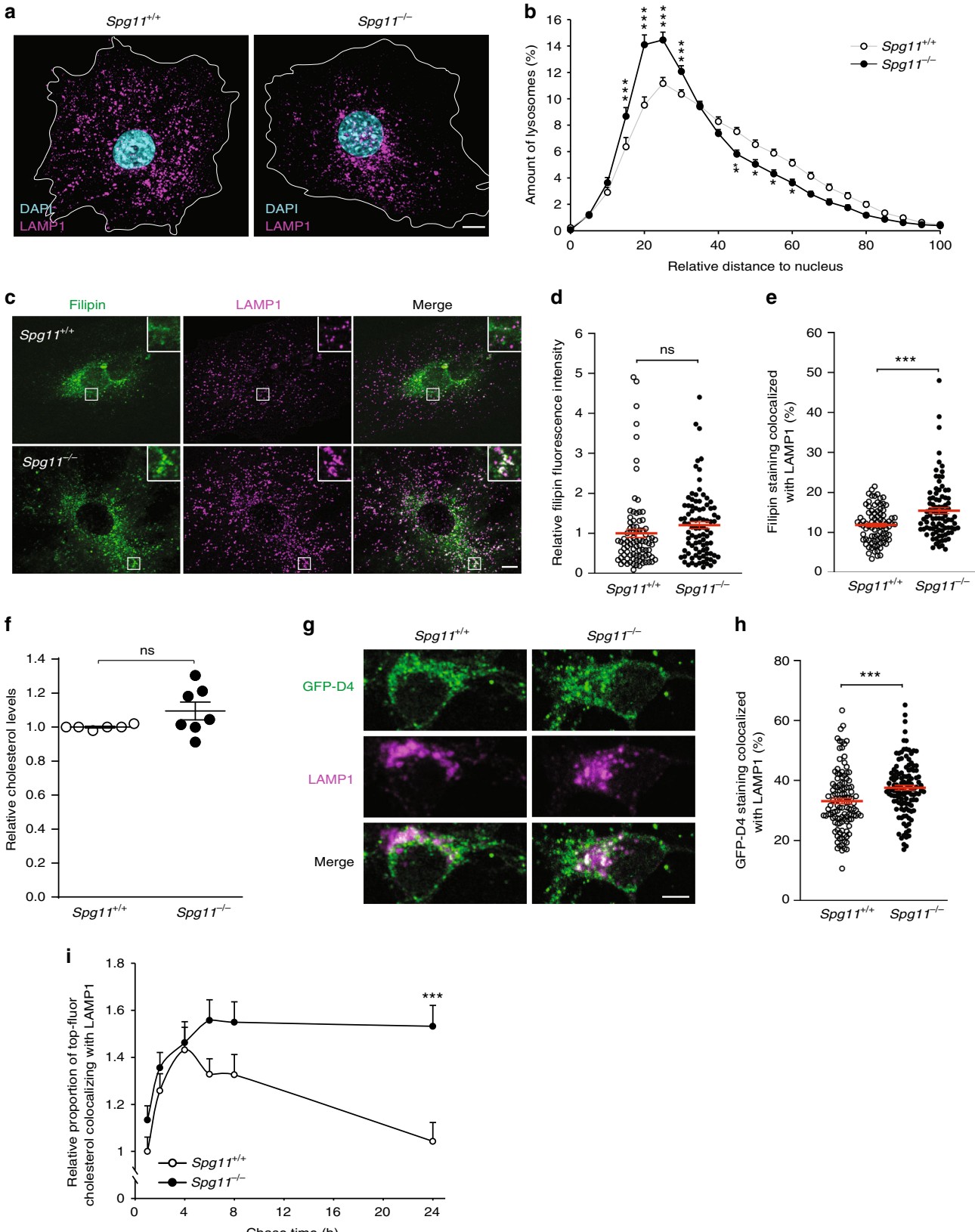

inhibition of dynamin by dynasore increased the proportion of cholesterol colocalized with late endosomes/lysosomes in control, but not in *Spg11*[−/−] fibroblasts (Fig. 2d). These data suggest that spatacsin and dynamin cooperate in a same pathway to clear cholesterol from late endosomes/lysosomes. Overall, these data

suggest that the formation of tubules on lysosomes contributes to the clearance of cholesterol from lysosomes.

We investigated whether lysosomal tubules are used for cholesterol trafficking by transfecting fibroblasts with a vector expressing LAMP1-mCherry and incubating them with LDL

**Fig. 1** The loss of spatacsin ($Spg11^{-/-}$) promotes the accumulation of cholesterol in late endosomes/lysosomes. **a** Immunostaining of $Spg11^{+/+}$ and $Spg11^{-/-}$ fibroblasts with the late endosome/lysosome marker LAMP1. Nuclei are stained with DAPI. White lines indicate the cell periphery. Scale bar: 10 µm. **b** Distribution of late endosomes/lysosomes in $Spg11^{+/+}$ and $Spg11^{-/-}$ fibroblasts. The maximum distance between particles and the nucleus was fixed at 100 for each cell. Late endosomes/lysosomes cluster more around the nuclei of $Spg11^{-/-}$ than $Spg11^{+/+}$ fibroblasts. The graph shows the mean ± SEM. $N = 65$ cells from three independent experiments. Two-way ANOVA: ***$p < 0.0001$; **$p < 0.01$; *$p < 0.05$. **c** Staining of cholesterol with filipin and late endosomes/lysosomes by the marker LAMP1 in $Spg11^{+/+}$ and $Spg11^{-/-}$ fibroblasts. Insets show a higher magnification of the zone highlighted by a white square. Scale bar: 10 µm. **d** Quantification of the intensity of filipin staining of whole cells showing no significant difference in the total amount of cholesterol in $Spg11^{+/+}$ and $Spg11^{-/-}$ fibroblasts. The graph shows the mean ± SEM. $N > 85$ cells from three independent experiments. $T$-test: $p = 0.83$. **e** Quantification of the amount of filipin staining colocalized with the marker LAMP1, showing more cholesterol in late endosomes/lysosomes in $Spg11^{-/-}$ than $Spg11^{+/+}$ fibroblasts. The graph shows the mean ± SEM. $N > 85$ cells from three independent experiments. $T$-test: ***$p < 0.0001$. **f** Biochemical quantification of total cholesterol levels in $Spg11^{+/+}$ ($N = 7$) and $Spg11^{-/-}$ ($N = 6$) neurons. The graph shows the mean ± SD. Mann–Whitney test: $p = 0.63$. **g** Staining of cholesterol with GFP-D4 probe and immunostaining of the late endosome/lysosome marker LAMP1 in $Spg11^{+/+}$ and $Spg11^{-/-}$ primary cortical neurons. Scale bar: 5 µm. **h** Quantification of the amount of GFP-D4 staining colocalized with the marker LAMP1, showing more cholesterol in late endosomes/lysosomes in $Spg11^{-/-}$ than $Spg11^{+/+}$ neurons. The graph shows the mean ± SEM. $N > 110$ cells from three independent experiments. $T$-test: ***$p < 0.001$. **i** Quantification of the amount of Top-Fluor cholesterol colocalized with the marker LAMP1 in $Spg11^{+/+}$ and $Spg11^{-/-}$ fibroblasts over time. The graph shows the mean ± SEM. $N > 95$ cells analyzed in three independent experiments. Two-way ANOVA: ***$p < 0.0001$

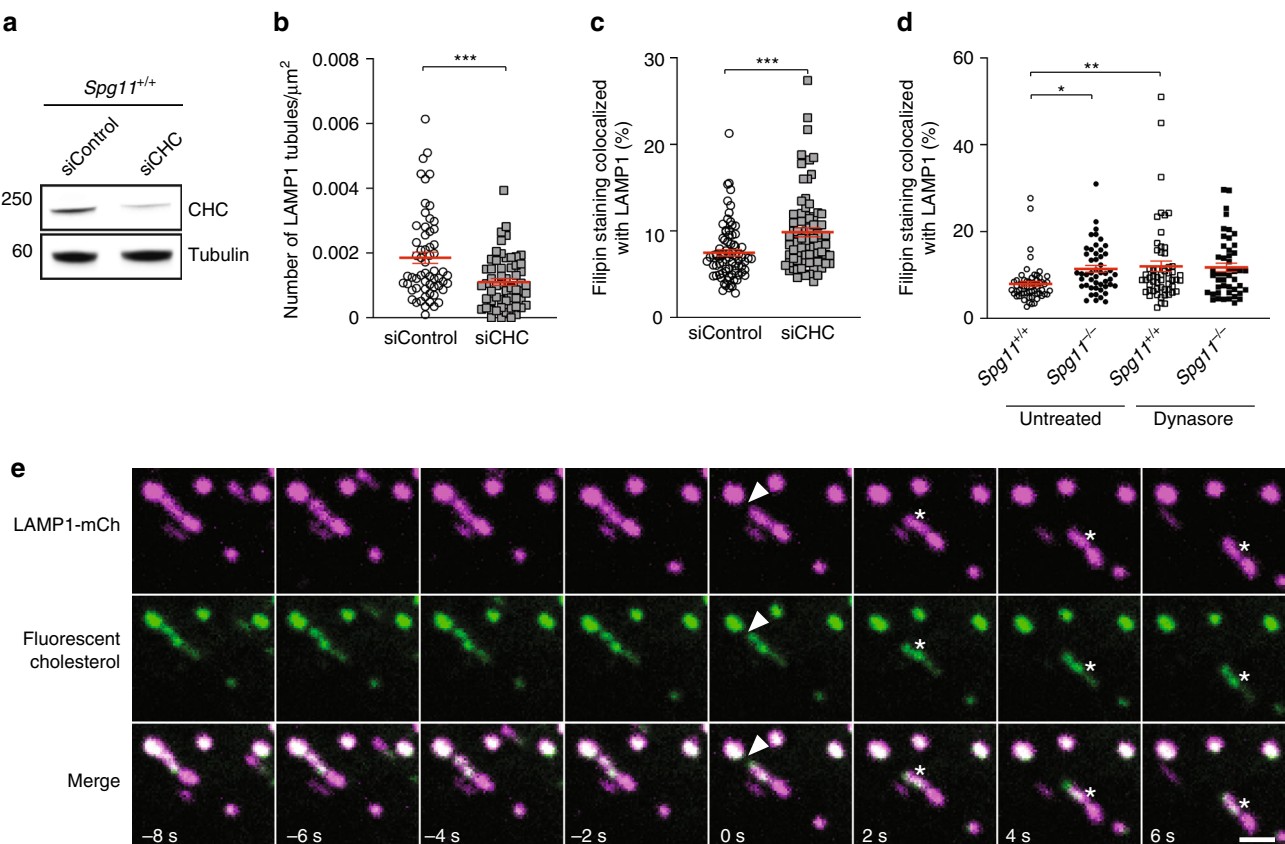

**Fig. 2** Inhibition of tubule formation in late endososmes/lysosomes causes the accumulation of cholesterol. **a** Western blot showing the downregulation of clathrin heavy chain (CHC) in wild-type mouse embryonic fibroblasts transfected with siRNA targeting CHC. **b** Quantification of the number of LAMP1-positive tubules in wild-type fibroblasts transfected with a control siRNA or a siRNA that downregulates CHC and expressing LAMP1 fused to mCherry, analyzed by live imaging. The graph shows the mean ± SEM. $N > 58$ cells analyzed in three independent experiments. $T$-test: ***$p = 0.0004$. **c** Quantification of the amount of filipin staining colocalized with the LAMP1 marker in fibroblasts transfected with a control siRNA or a siRNA that downregulates CHC. Downregulation of CHC resulted in a higher amount of cholesterol in late endosomes/lysosomes. The graph shows the mean ± SEM. $N > 78$ cells analyzed in three independent experiments. $T$-test: ***$p = 0.0002$. **d** Two-hour treatment of fibroblasts with the dynamin inhibitor dynasore (40 µM) induces the accumulation of cholesterol in $Spg11^{+/+}$ but not $Spg11^{-/-}$ fibroblasts. The graph shows the mean ± SEM. $N > 78$ cells analyzed in three independent experiments. Two-way ANOVA: *$p = 0.037$, **$p = 0.0098$. **e** Live imaging of fibroblasts expressing LAMP1-mCherry and loaded with fluorescent cholesterol coupled to LDL. Note the presence of fluorescent cholesterol in tubules emanating from LAMP1-positive late endosomes/lysosomes (asterisk). Arrowheads point to a lysosomal tubule undergoing fission. Scale bar: 2 µm

loaded with fluorescent cholesterol for 2 h in the presence of U18666a. This compound promotes the strong accumulation of cholesterol in late endosomes and lysosomes[22]. Twenty minutes after U18666a washout, which allows cholesterol egress from lysosomes, live imaging showed the fluorescent cholesterol to be localized to lysosomal tubules (Fig. 2e). Occasionally, tubules fission gave rise to new vesicles containing cholesterol (Fig. 2e), suggesting that tubulation in late endosomes/lysosomes is involved in cholesterol trafficking.

**Lysosome tubulation regulates plasma membrane cholesterol.** In cells, cholesterol levels are high in the plasma membrane, intermediate in late endosome/lysosomes, and low in the endoplasmic reticulum (ER)[23]. We investigated whether the accumulation of cholesterol in late endosomes/lysosomes changes its concentration in the plasma membrane by staining cholesterol in the plasma membrane of live cells using the probe GFP-D4. Cholesterol levels in the plasma membrane were significantly lower in $Spg11^{-/-}$ than control cells (Fig. 3a, b). We confirmed this result by determining total and plasma membrane cholesterol levels by an enzymatic assay. The total amount of cholesterol was the same in $Spg11^{-/-}$ and $Spg11^{+/+}$ cells (Supplementary Fig. 1b), but it was lower in the plasma membrane of $Spg11^{-/-}$ than $Spg11^{+/+}$ cells (Fig. 3c). Similarly, the inhibition of tubule formation in late endosomes/lysosomes by downregulation of clathrin heavy chain or dynasore treatment led to the accumulation of cholesterol in late endosomes/lysosomes at the expense of the plasma membrane (Figs. 2 and 3d, e). Overall, these results show that impaired trafficking of cholesterol out of late endosomes/lysosomes due to alterations in the formation of tubules results in decreased levels of cholesterol in the plasma membrane.

**Depletion of plasma membrane cholesterol increases store-operated calcium entry.** We then investigated the consequences of impaired trafficking of cholesterol from lysosomes to the plasma membrane by analyzing cells deficient in spatacsin, which is required for the initiation of tubule formation[11]. On electron microscopy preparations, the loss of spatacsin significantly increased the number and size of the contacts between ER and the plasma membrane (Fig. 4a–c). Such close contacts play a role in various cellular functions and notably regulate transfer of lipids between the membranes, or homeostasis of calcium[24,25]. Upon depletion of the intracellular calcium store of the ER, the ER calcium sensor STIM1 oligomerizes and interacts with the plasma

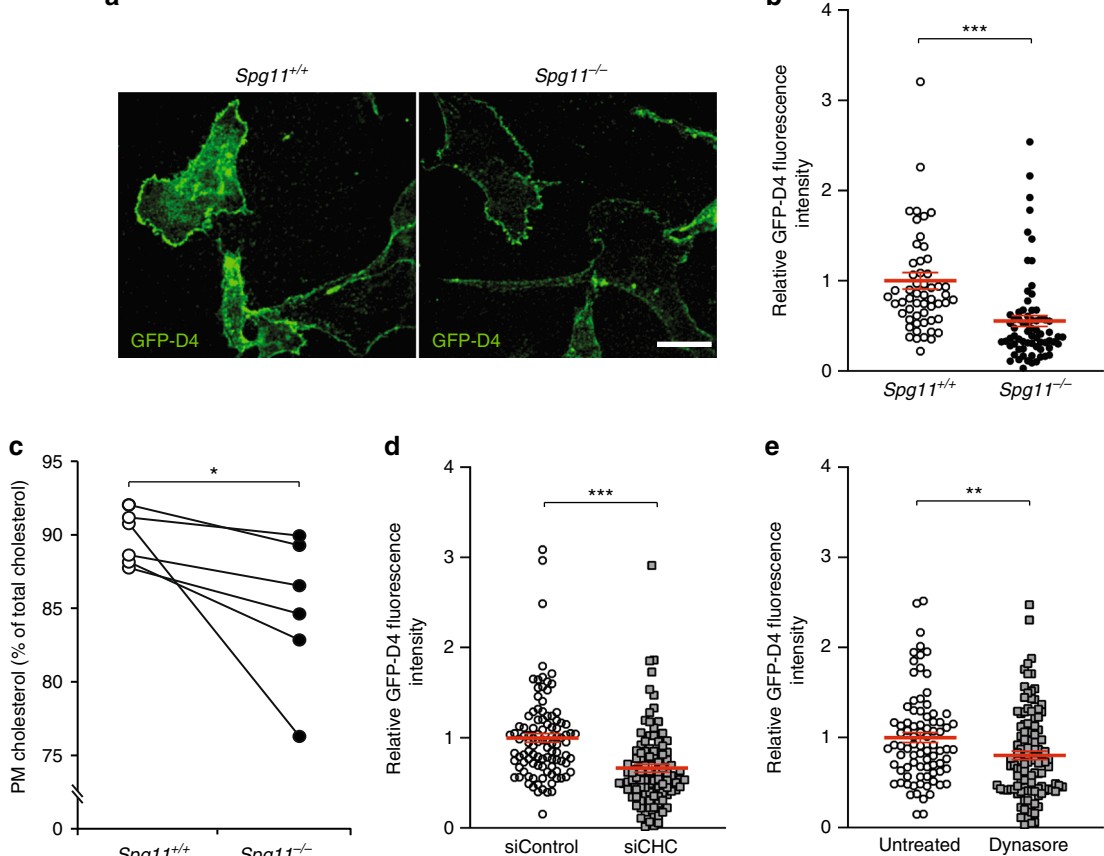

**Fig. 3** The inhibition of tubule formation on late endosomes/lysosomes lowers cholesterol content in the plasma membrane. **a**. Staining of live fibroblasts with the probe GFP-D4, which allows staining of the plasma membrane cholesterol only. Scale bar: 10 μm. **b** Quantification of the intensity of GFP-D4 staining performed on live $Spg11^{+/+}$ and $Spg11^{-/-}$ fibroblasts, showing a lower level of plasma membrane cholesterol in $Spg11^{-/-}$ than $Spg11^{+/+}$ fibroblasts. The graphs show the mean ± SEM. $N > 95$ cells analyzed in at least three independent experiments. $T$-test: ***$p < 0.0001$. **c** Biochemical quantification of the proportion of cholesterol present in the plasma membrane in $Spg11^{+/+}$ and $Spg11^{-/-}$ fibroblasts, showing a lower level of plasma membrane cholesterol in $Spg11^{-/-}$ than $Spg11^{+/+}$ fibroblasts. $N = 6$ independent assays. Wilcoxon paired test: *$p = 0.031$. **d** Quantification of the intensity of GFP-D4 staining performed on live control fibroblasts transfected with control siRNA or siRNA targeting CHC. Downregulation of CHC decreases the amount of cholesterol in the plasma membrane. The graph shows the mean ± SEM. $N > 100$ cells analyzed in two independent experiments. $T$-test: ***$p < 0.0001$. **e** Quantification of the intensity of GFP-D4 staining performed on live control fibroblasts treated with dynasore (40 μM, 2 h). Inhibition of dynamin decreases the amount of cholesterol in the plasma membrane. The graph shows the mean ± SEM. $N > 80$ cells analyzed in three independent experiments. $T$-test: **$p = 0.0062$

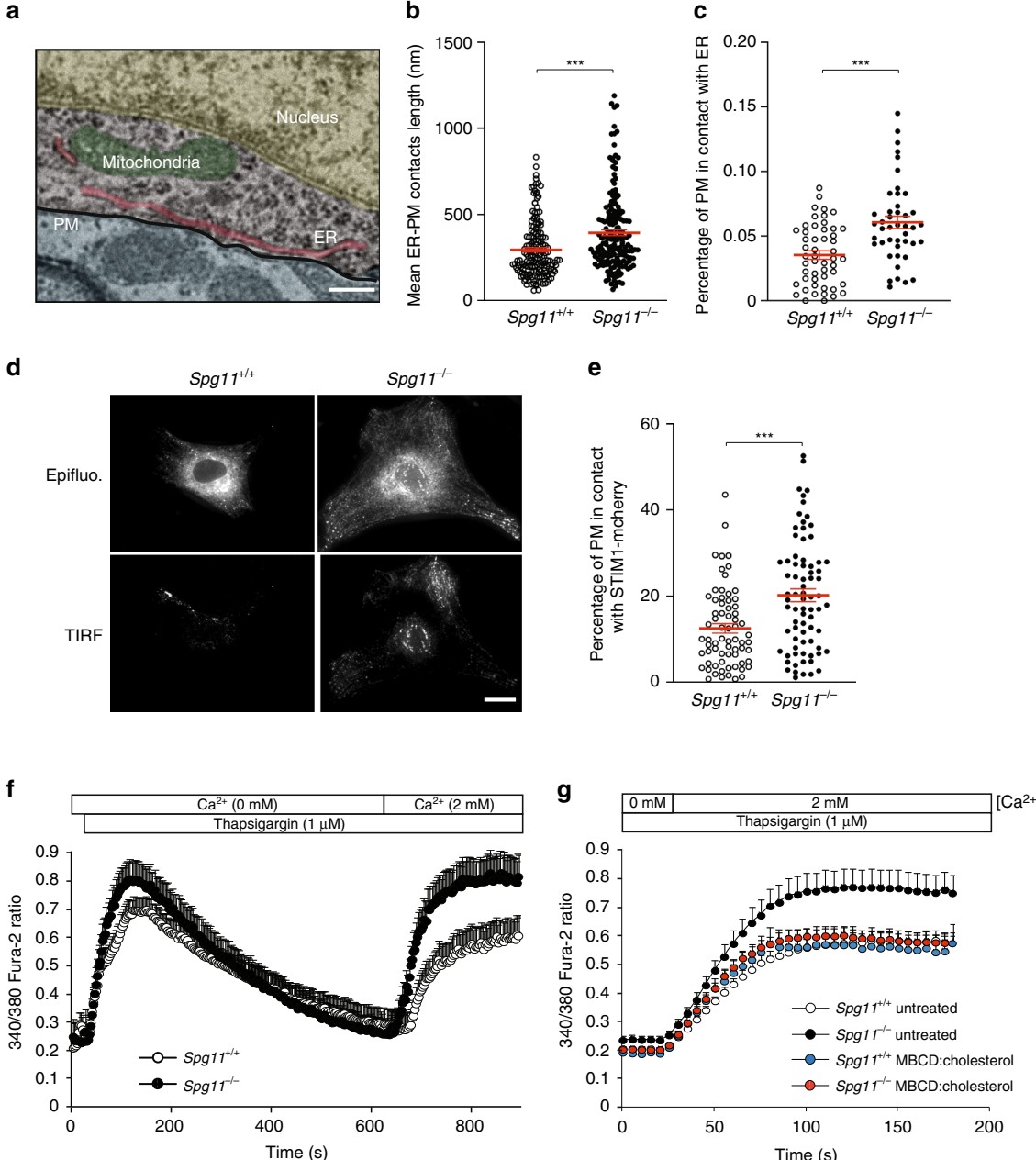

**Fig. 4** The depletion of plasma membrane cholesterol promotes higher store-operated calcium entry. **a** Electron micrograph of neurons in the cortex of a 2-month-old *Spg11*$^{-/-}$ mouse, showing close contact between the endoplasmic reticulum (ER) and plasma membrane (PM). False colors highlight the various cellular compartments. Scale bar: 250 nm. **b, c** Quantification of contacts between the ER and plasma membrane, defined as the zone where the distance between the two membranes is lower than 30 nm. **b** Quantification of the mean length of individual contacts between the ER and plasma membrane in the cortex of 2-month-old *Spg11*$^{-/-}$ or *Spg11*$^{+/+}$ mice. **c** Quantification of the percentage of the plasma membrane in close contact with the ER in the cortex of 2-month-old *Spg11*$^{-/-}$ or *Spg11*$^{+/+}$ mice. The graphs represent the mean ± SEM. $N > 23$ cells analyzed in two independent mice for each group. *T*-test: ***$p < 0.0001$. **d** *Spg11*$^{-/-}$ or *Spg11*$^{+/+}$ mouse embryonic fibroblasts transfected with vectors expressing STIM1-mCherry imaged by epifluorescence or total internal reflection microscopy (TIRF). Scale bar: 10 μm. **e** Quantification of the percentage of the cellular area with STIM1-mCherry staining detected by TIRF microscopy, indicating close contact between STIM1-mCherry and the plasma membrane. The graph shows the mean ± SEM. $N > 60$ cells from three independent experiments. *T*-test: ***$p < 0.0001$. **f** Evaluation of extracellular calcium import by SOCE. Cytosolic calcium was measured with Fura-2 in the absence of extracellular calcium. The ER calcium store was depleted with thapsigargin, 2 mM CaCl$_2$ added to the extracellular medium, and the increase in cytosolic calcium measured with Fura-2, allowing the quantification of SOCE. The graph shows the mean ± SEM. $N > 35$ cells from three independent experiments. **g** Increasing cholesterol levels in the plasma membrane with methyl-β-cyclodextrin (MBCD) loaded with cholesterol decreases store-operated calcium entry in *Spg11*$^{-/-}$ fibroblasts, measured by the addition of 2 mM extracellular calcium after a 10-min treatment with thapsigargin. The graph shows the mean ± SEM. $N > 60$ cells from three independent experiments

membrane calcium channel Orai1, forming close contacts between the ER and the plasma membrane and allowing the import of extracellular calcium to restore normal intracellular calcium homeostasis[25]. This mechanism is known as store-operated calcium entry (SOCE). We analyzed the proximity of the ER calcium sensor STIM1 and the plasma membrane by total internal reflection fluorescence (TIRF) in cells transfected with a vector expressing STIM1-mCherry. TIRF microscopy performed on fibroblasts under basal conditions confirmed that the proportion of the plasma membrane in close contact with the ER calcium sensor STIM1 was higher in $Spg11^{-/-}$ than control cells (Fig. 4d, e).

Levels of cholesterol in the plasma membrane regulate SOCE[26,27]. We therefore tested whether lower levels of cholesterol in the plasma membrane, caused by the loss of spatacsin, altered SOCE. We treated fibroblasts in $Ca^{2+}$-free medium with the SERCA inhibitor thapsigargin to deplete the ER calcium store and trigger SOCE. We then added 2 mM calcium in the extracellular medium and calcium import was measured using the cytosolic calcium probe Fura-2. $Spg11^{-/-}$ cells imported more extracellular calcium than $Spg11^{+/+}$ cells (Fig. 4f), suggesting that the loss of spatacsin promoted SOCE under basal conditions.

We then investigated whether the increased SOCE observed in the absence of spatacsin was due to lower levels of cholesterol in the plasma membrane. We increased plasma membrane cholesterol levels by exposing $Spg11^{-/-}$ fibroblasts for 1 h to methyl-β-cyclodextrin loaded with cholesterol (Supplementary Fig. 3a, b). This restored normal SOCE in $Spg11^{-/-}$ fibroblasts (Fig. 4g), suggesting that cholesterol depletion from the plasma membrane due to impaired lysosomal tubulation is responsible for the increase in SOCE when spatacsin function is lost.

**Plasma membrane cholesterol regulates cytosolic calcium levels.** SOCE promotes the entry of extracellular calcium into the cytosol that is normally taken up by the ER[25,28]. We monitored whether the increased SOCE due to the loss of spatacsin modified cytosolic calcium levels in resting cells. Cytosolic calcium levels were slightly, but significantly, higher in $Spg11^{-/-}$ than $Spg11^{+/+}$ fibroblasts (Fig. 5a). We tested whether this increase in cytosolic calcium was a consequence of increased SOCE by reducing extracellular calcium levels to 0.4 mM by adding EGTA to the culture medium for 1 h. Under these conditions, resting cytosolic calcium levels were significantly reduced in both $Spg11^{+/+}$ and $Spg11^{-/-}$ fibroblasts (Fig. 5a). We confirmed this result by downregulating the expression of STIM1 by transfecting fibroblasts with specific siRNA (Fig. 5b). Downregulation of STIM1 decreased SOCE and restored normal cytosolic calcium levels in $Spg11^{-/-}$ cells (Fig. 5b, c), demonstrating that enhanced SOCE increases cytosolic calcium levels in the absence of spatacsin. Finally, we restored normal cytosolic calcium levels when we increased cholesterol levels in the plasma membrane of $Spg11^{-/-}$ fibroblasts (Supplementary Fig. 3), suggesting that the increase in SOCE, caused by lower plasma membrane cholesterol levels, is responsible for the alteration of cytosolic calcium levels (Fig. 5d).

**Cytosolic calcium contributes to cholesterol accumulation in lysosomes.** Among other cellular functions, the entry of extracellular calcium by SOCE has been proposed to regulate the nuclear translocation of TFEB[29], which is a major regulator of lysosome function[30]. We monitored the amount of nuclear TFEB, which represents the transcriptionally active protein[30], in $Spg11^{-/-}$ and $Spg11^{+/+}$ fibroblasts. The amount of nuclear TFEB was significantly higher in $Spg11^{-/-}$ than $Spg11^{+/+}$ fibroblasts, whereas cytosolic levels of TFEB were not significantly different (Fig. 6a). Decreasing cytosolic calcium levels using the

intracellular chelator EGTA-AM or by lowering extracellular calcium levels decreased the amount of nuclear TFEB in $Spg11^{-/-}$ fibroblasts, suggesting that higher SOCE in $Spg11^{-/-}$ fibroblasts is responsible for the nuclear translocation of the transcription factor (Fig. 6a). Translocation of TFEB into the nucleus depends on its phosphorylation state[31], and it can be phosphorylated by mTOR. The levels of phosphorylated S6 protein and S6 kinase, two mTOR substrates, were similar in $Spg11^{-/-}$ and $Spg11^{+/+}$ fibroblasts (Supplementary Fig. 4a), suggesting that mTOR activity is not altered in absence of spatacsin and that it is not responsible for nuclear TFEB in $Spg11^{-/-}$ fibroblasts. We then examined whether the entry of calcium mediated by SOCE in $Spg11^{-/-}$ fibroblasts could promote the nuclear translocation of TFEB by regulating the calcium-dependent phosphatase calcineurin[32]. The amount of nuclear TFEB was partially restored in $Spg11^{-/-}$ fibroblasts upon transfection with a siRNA downregulating calcineurin compared to a control siRNA (Fig. 6b, c). Together, these data suggest that entry of calcium by SOCE in $Spg11^{-/-}$ fibroblasts mediates the calcium-dependent dephosphorylation of TFEB, allowing its nuclear translocation.

Since TFEB regulates many lysosome functions, we wondered whether the higher levels of nuclear TFEB due to higher cytosolic calcium levels could regulate the formation of tubules of late endosomes/lysosomes and modulate the cholesterol content in this compartment. We decreased SOCE by downregulating STIM1 or reducing extracellular free $Ca^{2+}$ levels. These treatments partially restored tubule formation in the absence of spatacsin (Fig. 7a and Supplementary Fig. 4b). Similarly, treatment with the intracellular calcium chelator EGTA-AM, to decrease cytosolic calcium levels, increased the number of lysosomes with tubules in $Spg11^{-/-}$ fibroblasts (Supplementary Fig. 4c). We tested whether these effects where mediated by TFEB by downregulating its expression using siRNA, leading to lower levels of TFEB in both the cytoplasm and nucleus (Fig. 7b). Downregulation of TFEB in $Spg11^{-/-}$ fibroblasts partially restored the number of lysosomes with tubules (Fig. 7c). Overall, these data suggest that altered calcium homeostasis impairs the formation of tubules on lysosomes in the absence of spatacsin in a TFEB-dependent manner.

We showed that tubule formation is required for the clearance of cholesterol from lysosomes (Fig. 1). We thus investigated whether treatment that restores the formation of tubules in the absence of spatacsin also has an effect on cholesterol accumulation in late endosomes/lysosomes. Decreasing SOCE by downregulating STIM1 expression corrected the accumulation of cholesterol observed in lysosomes in $Spg11^{-/-}$ fibroblasts (Fig. 7d). Similarly, decreasing cytosolic calcium levels with EGTA-AM decreased cholesterol levels in late endosomes/lysosomes in $Spg11^{-/-}$ fibroblasts (Fig. 7e) and $Spg11^{-/-}$ neurons (Supplementary Fig. 4d). Downregulation of TFEB in $Spg11^{-/-}$ fibroblasts also decreased the proportion of cholesterol in late endosomes/lysosomes (Fig. 7f). Since TFEB was shown to regulate lipid metabolism in liver[33], we monitored whether downregulation of TFEB could activate the transcription factor SREBP that regulates cholesterol synthesis[34]. SREBP is activated by its cleavage, and we detected no change in the levels of activated SREBP between $Spg11^{+/+}$ and $Spg11^{-/-}$ fibroblasts, whether TFEB was downregulated or not (Supplementary Fig. 4e). Together, these data suggest that increased cytosolic calcium levels contributed to the accumulation of cholesterol in a TFEB-dependent manner.

We also showed that the accumulation of cholesterol in late endosomes/lysosomes slightly decreases cholesterol levels in the plasma membrane (Fig. 3). We reasoned the treatment that restores the distribution of cholesterol in late endosomes/lysosomes in $Spg11^{-/-}$ fibroblasts should also restore normal

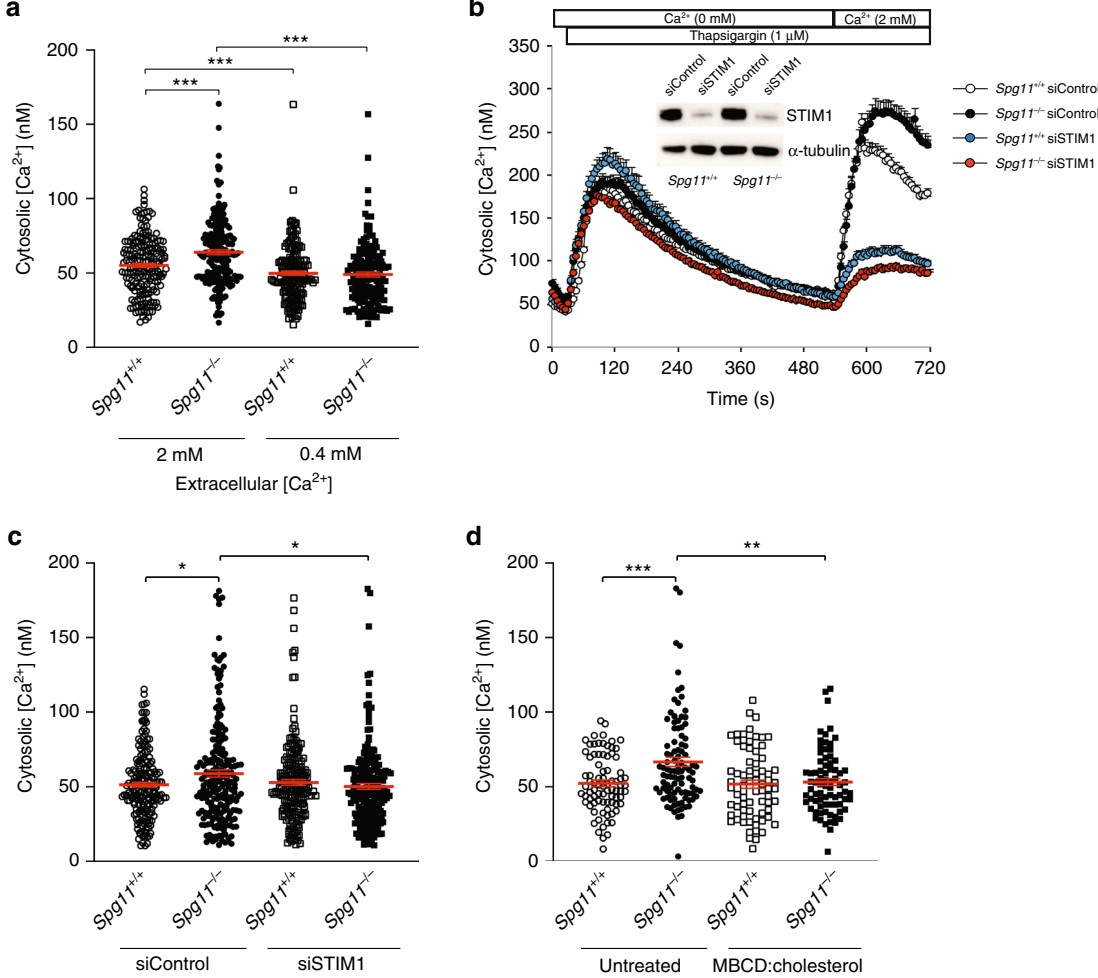

**Fig. 5 High store-operated calcium entry in the absence of spatacsin increases cytoplasmic calcium levels. a** Quantification of cytosolic calcium levels in *Spg11*$^{+/+}$ and *Spg11*$^{-/-}$ fibroblasts in normal medium or medium supplemented with EGTA to lower the extracellular calcium to 0.4 mM. The graphs represent the mean ± SEM. *N* > 159 cells from three independent experiments. Two-way ANOVA: ***$p$ < 0.0001. **b** Downregulation of STIM1 strongly abrogates store-operated calcium entry in *Spg11*$^{+/+}$ and *Spg11*$^{-/-}$ fibroblasts. The graphs show the mean ± SEM. *N* > 55 cells from three independent experiments. Insert: western blot showing the downregulation of STIM1 in *Spg11*$^{+/+}$ and *Spg11*$^{-/-}$ fibroblasts transfected with siRNA directed against STIM1. **c** Downregulation of STIM1 decreases the levels of cytosolic calcium in *Spg11*$^{-/-}$ fibroblasts to those measured in *Spg11*$^{+/+}$ fibroblasts. The graph shows the mean ± SEM. *N* > 190 cells analyzed in three independent experiments. Two-way ANOVA: *$p$ < 0.05. **d** Treatment of *Spg11*$^{+/+}$ or *Spg11*$^{-/-}$ fibroblasts with methyl-β-cyclodextrin (MBCD) loaded with cholesterol for 1 h restores normal cytosolic calcium levels in *Spg11*$^{-/-}$ cells. The graph shows the mean ± SEM. *N* > 70 cells from three independent experiments. Two-way ANOVA: **$p$ = 0.0017, ***$p$ = 0.0006

levels of cholesterol in the plasma membrane. Inhibiting SOCE via STIM1 downregulation indeed corrected cholesterol levels in the plasma membrane of *Spg11*$^{-/-}$ fibroblasts (Fig. 7g), showing that dysregulation of calcium homeostasis contributed to the observed alterations in cholesterol trafficking. This demonstrates that impaired calcium homeostasis due to the accumulation of cholesterol in late endosomes/lysosomes contributed to the maintenance or enhancement of the imbalanced cholesterol distribution.

## Discussion

Loss of spatacsin leads to accumulation of lipids in lysosomes, both in neuronal and non-neuronal cells[4], but the mechanisms underlying the accumulation of lipids in this compartment are unknown. Here we show that spatacsin is implicated in the trafficking of cholesterol and demonstrate that alteration of this trafficking pathway has functional consequences for the plasma membrane and calcium homeostasis, affecting lysosome function.

Cholesterol is an essential constituent of cellular membranes, but is unevenly distributed within subcellular compartments[23,35]. The lipid composition of membranes, including the amount of cholesterol, affects their biological functions[36]. The mechanisms that regulate cholesterol transport between subcellular compartments thus appear to be critical for cellular functions[14]. The transport of cholesterol out of lysosomes requires the proteins Niemann Pick Type C (NPC) 1 and 2 that likely allow cholesterol to be integrated in the lysosomal membrane[14,23,37]. However, the dissection of mechanisms allowing cholesterol transport is complicated by the co-existence of vesicular transport of cholesterol[21] and non-vesicular trafficking of cholesterol at the levels of contact sites between lysosomes and other subcellular compartments[35,38].

The formation of tubules on lysosomes requires clathrin, spatacsin, and dynamin. These proteins are involved in the recycling of lysosome membranes after the termination of autophagy[9–11]. Although autophagic lysosome recovery occurs after the termination of autophagy, we show here that this machinery is also used to clear cholesterol from late endosomes/lysosomes by

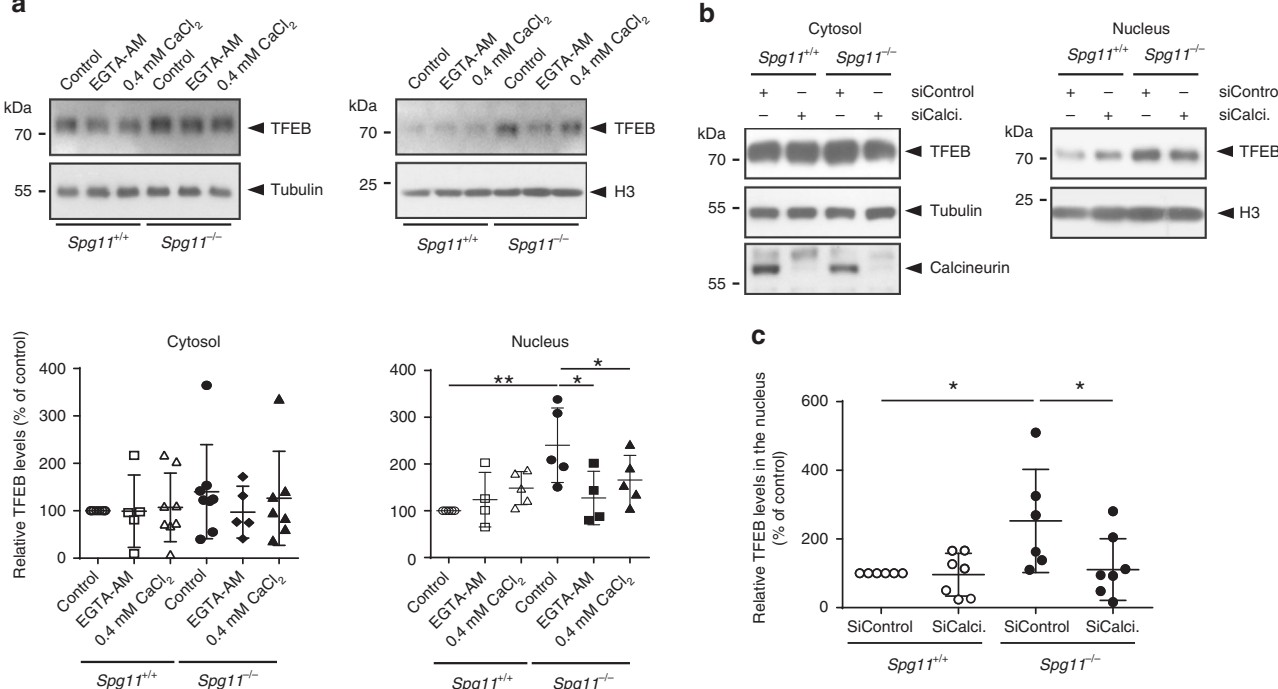

**Fig. 6 High cytosolic calcium levels promotes nuclear translocation of TFEB in the absence of spatacsin. a** Western blot of TFEB in cytosolic and nuclear fractions of $Spg11^{+/+}$ and $Spg11^{-/-}$ fibroblasts cultured for 2 h in normal medium or medium containing either 0.4 mM $CaCl_2$ or 0.5 μM EGTA-AM. Graphs show the quantification of the amount of TFEB normalized to the levels of α-tubulin (Cytosol) and Histone H3 (Nuclei). One-way ANOVA: *$p < 0.05$, **$p < 0.01$. **b** Western blots of TFEB in cytosolic and nuclear fractions of $Spg11^{+/+}$ and $Spg11^{-/-}$ fibroblasts transfected with control siRNA or a specific siRNA that downregulates calcineurin (Calci). Downregulation of calcineurin is evidenced by western blot in the cytosolic fraction. **c** Quantification of the amount of nuclear TFEB normalized to the levels of Histone H3 upon downregulation of calcineurin (SiCalci). One-way ANOVA: *$p < 0.05$

tubulation under basal conditions. Accordingly, downregulation of spatacsin was shown to decrease the formation of tubules on late endosomes/lysosomes under basal conditions[11]. The accumulation of cholesterol in late endosomes/lysosomes due to the inhibition of tubulation leads to lower cholesterol levels in the plasma membrane. shRNA screening consistently identified spatacsin as a regulator of cholesterol trafficking from lysosomes toward the plasma membrane[39]. The formation of tubules could give rise to vesicles that may participate in the vesicular trafficking of cholesterol from late endosomes/lysosomes to the plasma membrane. The mechanism that regulates such trafficking is not clear, but it may involve Rab8a and myosin5b, as previously observed[21].

Changes in the concentration of cholesterol in the plasma membrane enhance the entry of extracellular calcium by SOCE and leads to higher cytosolic calcium levels, which could contribute to alter calcium signaling[17]. Cholesterol affects SOCE in various cellular systems[26,40,41]. Global depletion of cholesterol in cells was shown to decrease SOCE[40,41]. In contrast, cholesterol depletion in the plasma membrane was shown to enhance SOCE[26], consistent with our observation that SOCE was higher when plasma membrane cholesterol levels were lower in absence of spatacsin. This effect could be mediated by the interaction of plasma membrane cholesterol with Orai1 channel, regulating its activity[26]. A recent study showed that the entry of calcium by SOCE promotes nuclear translocation of the master lysosomal gene TFEB, promoting its transcriptional activity[29] and thereby regulating autophagy, lysosome biogenesis, and metabolism of lipids[33]. In accordance with these results, we observed increased nuclear translocation of TFEB in absence of spatacsin. Decreasing calcium entry or cytosolic calcium levels was sufficient to restore normal nuclear TFEB

levels in the absence of spatacsin. Thus, changes in plasma membrane composition could indirectly modulate lysosomal function through calcium-dependent regulation of TFEB. Nuclear translocation of TFEB depends on its phosphorylation state, and the calcium-dependent phosphatase calcineurin was shown to dephosphorylate TFEB allowing its nuclear translocation[32]. Our data suggest that calcium entry by SOCE allows calcineurin-dependent nuclear translocation of TFEB. Major kinases responsible for TFEB phosphorylation include mTOR, ERK, GSK3β, and AKT[31]. Loss of spatacsin has been shown to impair GSK3β phosphorylation[42], and this signaling pathway could also contribute to the higher nuclear translocation of TFEB in $Spg11^{-/-}$ cells.

TFEB activation has been proposed to promote cellular clearance in several lysosomal storage disorders[43]. It could be hypothesized that increased nuclear translocation of TFEB is a compensatory mechanism to restore lysosomal function in $Spg11^{-/-}$ cells. However, downregulation of TFEB or treatments that compensated the nuclear translocation of TFEB in $Spg11^{-/-}$ cells partially restored the formation of tubules on late endosomes/lysosomes, in the absence of spatacsin, and restored cholesterol homeostasis. These data therefore suggest that nuclear translocation of TFEB inhibited the formation of tubules and the clearance of cholesterol in lysosomes. Nuclear translocation of TFEB may induce the expression of proteins that block the tubulation of lysosomes and the recycling of lysosomal membrane, but the nature of such factors is still to be uncovered.

The interdependence of cholesterol trafficking and calcium homeostasis that we observed highlights a homeostatic equilibrium in which the impairment of cholesterol clearance from lysosomes modifies plasma membrane composition, thus affecting calcium homeostasis and lysosomal cholesterol content in a

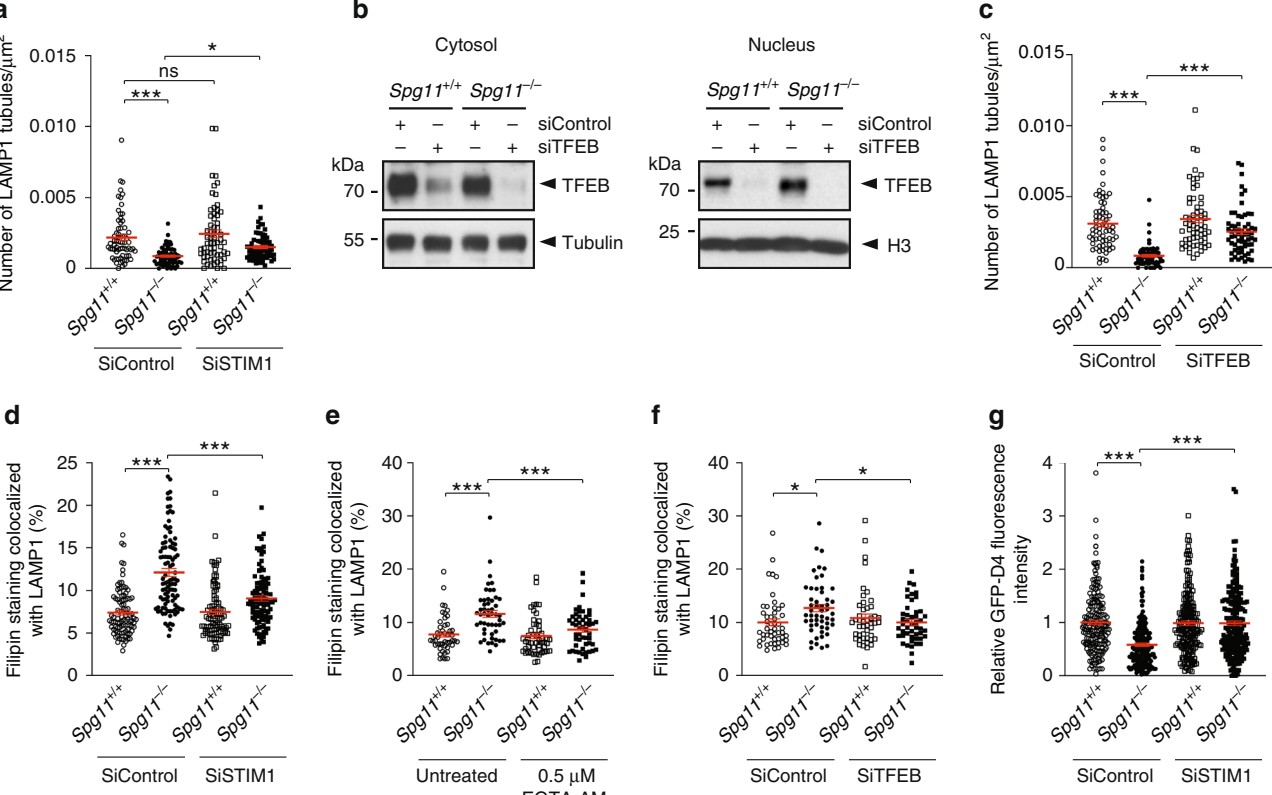

**Fig. 7** High cytosolic calcium levels cause accumulation of cholesterol in late endosomes/lysosomes in the absence of spatacsin. **a** Quantification of the number of LAMP1-positive tubules in $Spg11^{+/+}$ and $Spg11^{-/-}$ fibroblasts expressing LAMP1-mCherry, analyzed by live imaging. The graphs show the mean ± SEM. $N > 60$ cells analyzed in four independent experiments. Two-way ANOVA: $*p = 0.034$, $***p < 0.0001$. **b** Western blots of TFEB in cytosolic and nuclear fractions of $Spg11^{+/+}$ and $Spg11^{-/-}$ fibroblasts transfected with control siRNA or a specific siRNA that downregulates TFEB. **c** Quantification of the number of LAMP1-positive tubules in $Spg11^{+/+}$ and $Spg11^{-/-}$ fibroblasts transfected with control siRNA or siRNA that downregulates TFEB. The graph shows the mean ± SEM. $N > 60$ cells analyzed in four independent experiments. Two-way ANOVA: $***p < 0.0001$. **d** Downregulation of STIM1 decreases the amount of cholesterol colocalized with LAMP1 in $Spg11^{-/-}$ fibroblasts. The graph shows the mean ± SEM. $N > 95$ cells analyzed in three independent experiments. Two-way ANOVA: $***p < 0.001$. **e** Lowering intracellular calcium levels with EGTA-AM (1 h) decreases the amount of cholesterol colocalized with LAMP1 in $Spg11^{-/-}$ fibroblasts. The graph shows the mean ± SEM. $N > 45$ cells analyzed in three independent experiments. Two-way ANOVA: $***p < 0.001$. **f** Downregulation of TFEB decreases the amount of cholesterol colocalized with LAMP1 in $Spg11^{-/-}$ fibroblasts. The graph shows the mean ± SEM. $N > 45$ cells analyzed in three independent experiments. Two-way ANOVA: $*p < 0.05$. **g** Quantification of plasma membrane cholesterol with the probe GFP-D4, performed on live $Spg11^{+/+}$ and $Spg11^{-/-}$ fibroblasts, showing that downregulation of STIM1 restores normal levels of cholesterol in the plasma membrane in $Spg11^{-/-}$ fibroblasts. The graph shows the mean ± SEM. $N > 180$ cells analyzed in three independent experiments. Two-way ANOVA: $***p < 0.0001$.

TFEB-dependent manner. The compensatory role of the down-regulation of TFEB or the decrease in cytosolic calcium levels on the formation of tubules in late endosomes/lysosomes suggest that spatacsin could be indirectly implicated in the formation of tubules or that alternative mechanisms could compensate for the absence of spatacsin. However, the exact role of spatacsin in the maintenance of the homeostasis of calcium and cholesterol still need to be elucidated.

Our data support the hypothesis that the loss of spatacsin leads to similar impairment of cholesterol and calcium homeostasis both in non-neuronal cells and in neurons. Hereditary spastic paraplegia SPG11 is characterized by neuronal death in various brain regions[4,44]. The persistent deregulation of cholesterol distribution could lead to a slight modification of calcium homeostasis. Calcium plays a central role in cellular physiology and neuronal transmission[45] and a persistent change in cytosolic calcium levels could explain the behavioral alterations that were observed in $Spg11^{-/-}$ mice long before neurodegeneration occurred[4]. Alternatively, alteration in calcium homeostasis in

absence of spatacsin, could also contribute, in the long term, to neurodegeneration[46].

In conclusion, we demonstrate that loss of spatacsin function impairs trafficking of cholesterol leading to a strong alteration of cellular homeostasis that could contribute to neuronal dysfunction. Since SPG15 patients are indistinguishable from SPG11 patients[47], it would be interesting to investigate whether similar phenotype are observed in absence of spastizin. Atlastin that is mutated in the SPG3 form of HSP has also been proposed to modulate SOCE and lipid metabolism[48]. It would be interesting to investigate the role of atlastin in the distribution of cholesterol. Conversely, alterations of cholesterol trafficking in endosomes and lysosomes have also been described in models of Alzheimer's disease[49], and impaired distribution of cholesterol seems to play a crucial role in neurodegeneration in the case of Alzheimer's disease[50]. It may be informative to test whether the deregulation of cholesterol homeostasis in late endosomes in this disease also induces an alteration of cellular homeostasis that could contribute to persistent and deleterious impairment of lysosomal function.

## Methods

**Antibodies and chemicals**. Thapsigargin, filipin, CaCl$_2$, EDTA, and cholesterol were purchased from Sigma. EGTA-AM was purchased from Thermo Scientific. Dynasore was purchased from Abcam. Miglustat was purchased from Tocris. Antibodies used in the study were: mouse anti-α-tubulin (Abcam); rat anti-LAMP1 (Clone 1D4B, Development Studies Hybridoma Bank), mouse anti clathrin heavy chain (BD Biosciences), mouse anti-STIM1 (Cell Signalling), rabbit anti-TFEB (Proteintech), rabbit anti-calcineurin (Abcam), rabbit anti-histone H3 (Cell Signalling), and rabbit anti SREBP (Abcam). For immunoblotting, the secondary antibodies were conjugated to horseradish peroxidase (Jackson Laboratories) or fluorochromes (IR-dye 800 or IR-dye 680; LI-COR). Secondary antibodies used for immunofluorescence were purchased from Thermo Scientific.

**Mouse embryonic fibroblast cultures**. Spg11$^{-/-}$ mice in C57BL/6 N background were described previously[4]. Mouse embryonic fibroblasts (MEFs) were prepared using E14.5 embryos obtained from the breeding of heterozygous mice. After removing the head and inner organs, the body was minced with a razor blade and incubated in 0.25% trypsin/EDTA (Gibco) for 15 min at 37 °C. Cells were dissociated and grown in DMEM medium (Gibco) supplemented with 10% FBS and 1% penicillin/streptomycin. All experiments were performed between passages 4 and 6. At least three independent preparations of fibroblasts were used for each experiment.

**Primary cultures of cortical neurons**. Cortices of E14.5 embryos were mechanically dissociated in HBSS medium and plated at 25,000 neurons cm$^{-2}$ on poly-D-Lysine (250 μg ml$^{-1}$) coated glass coverslips. The neurons were grown in Neurobasal medium supplemented with 2% B27 (Gibco), 2 mM L-glutamine and 2% fetal bovine serum. Half of the medium was changed every 2 days and neurons were fixed after 6 days in vitro with 4% paraformaldehyde (PFA). When required, neurons were treated with miglustat (100 μM) from the second day after plating.

**Electron microscopy**. Two-month-old male and female Spg11$^{+/+}$ and Spg11$^{-/-}$ mice were anaesthetized and killed by intracardiac perfusion with a solution of 4% PFA in 0.1 M phosphate buffer at pH 7.4. Samples from the frontal cortex were fixed in 1% glutaraldehyde in the same buffer, post-fixed in 2% osmium tetroxide, dehydrated, and embedded in Araldite. Ultrathin sections were stained with uranyl acetate and lead citrate and examined using a Hitashi transmission electron microscope. Images were analyzed using ImageJ.

**Calcium imaging**. Cells grown in Lab-Tek™ (Nunc) were washed with HCSS buffer (120 mM NaCl, 5.4 mM KCl, 0.8 mM MgCl$_2$, 15 mM glucose, and 20 mM Hepes [pH 7.4]) and incubated with 2.5 μM Fura-2-AM (Life Technologies) for 30 min at room temperature in the dark. Cells were washed with HCSS and incubated 15 min at room temperature to allow Fura-2 de-esterification. Images were recorded with a Nikon Eclipse Ti-E inverted microscope, with excitation of Fura-2-AM loaded cells alternately at 340 and 380 nm. Emission at 510 nm was recorded. Conversion of Fura-2 ratios into cytosolic calcium concentrations was performed as previously described[51].

**Plasmids and transfection**. LAMP1-mCherry was obtained from Addgene (#45147). STIM1-mcherry was obtained from R. Lewis[52]. Fibroblasts were transfected with the Neon transfection system (Invitrogen), according to the manufacturer's instructions, using the following parameters: 1350 V, 30 ms, and one pulse. For overexpression studies, we used 0.5 μg DNA per 50 × 10$^3$ cells and the analysis was performed 24 h after transfection. For silencing studies, 50 × 10$^3$ cells were transfected with 1 pmol siRNA (Invitrogen) and analyzed 48 h later. The sequence of siRNA targeting STIM1 was 5′-GCAAGGAUGUUAUAUUUGATT-3′, that targeting clathrin heavy chain, 5′-CAUUGUCUGUGAUCGGUUUTT-3′, that targeting TFEB, 5′-CAACCUAAUUGAGAGAAGATT-3′, and that targeting calcineurin, 5′-GGGUUUGGAUAGGAUCAAUTT-3′.

**Immunofluorescence**. After fixation with 4% PFA, cells were incubated with PBS containing 10 mM NH$_4$Cl for 10 min at 22 °C to quench autofluorescence. Cells were incubated with a solution of 5% BSA/ 0.1% Triton X-100 in PBS for 30 min at 22 °C and then with primary antibodies in 5% BSA/0.1% Triton X-100 in PBS overnight at 4 °C. After washing, the cells were incubated with the secondary antibodies for 45 min at room temperature and mounted in Prolong Gold reagent (Thermo Scientific). Images were acquired with a Zeiss upright microscope equipped with a Plan-APOCHROMAT objective (×63; NA: 1.4), allowing acquisition of optical section images (Apotome 2 microscope).

**Lysosome positioning**. The position of lysosomes was assessed using ImageJ and MATLAB software. Signals from the nucleus (DAPI) and lysosomes (LAMP1) from an optical section were acquired with an Apotome 2 microscope. The centroid of the nucleus was determined using the DAPI signal and centroids of each lysosome were determined as the pixel with the highest intensity for each LAMP1-positive vesicle. The distance between lysosome centroids and the nucleus centroid was calculated. The results were expressed as the relative distance to the nucleus with 100 being the distance between the nucleus and the farthest lysosome.

**Live-cell imaging**. The formation of tubules in late endosomes/lysosomes was followed by live imaging of cells expressing LAMP1-mcherry at 37 °C and 5% CO2 using a Leica DMi8 microscope equipped with a Yokogawa Confocal Spinning Disk module. Cells were chosen randomly, with the only criterion being LAMP1-mCherry levels sufficiently high to detect lysosomal tubules.

**Cholesterol staining**. Cells were fixed with 4% PFA for 30 min at 22 °C. They were then incubated with filipin (50 μg ml$^{-1}$) in PBS supplemented with 10% FBS for 2 h at room temperature in the dark, without prior permeabilization. Cells were then processed for immunostaining when required. Cholesterol levels were quantified as the mean gray value using ImageJ. Colocalization of cholesterol staining with lysosomes was quantified using ImageJ on randomly chosen images of cultured fibroblasts. First, we created a mask corresponding to LAMP1 staining using the automatic threshold in Image J. The mask was copied to the corresponding fluorescence image of cholesterol. We quantified the total intensity of cholesterol fluorescence in the lysosome mask and expressed it as the percentage of total cholesterol fluorescence in every cell. A preparation of domain D4 of prefringolysin O fused to GFP (GFP-D4) was produced and purified as previously described[53]. Labeling of total cholesterol was performed by incubating fixed and permeabilized cells with 20 μg ml$^{-1}$ recombinant GFP-D4 for 20 min at 22 °C. Cholesterol of the outer leaflet of the plasma membrane was labeled by incubating live cells for 15 min at 22 °C with 20 μg ml$^{-1}$ GFP-D4 diluted in PBS containing 2 mM CaCl$_2$ and 0.8 mM MgCl$_2$. Cells were then fixed with 4% PFA for 20 min and processed for imaging.

**Cholesterol measurement**. Cells cultured in 60 mm petri dishes were harvested and lysed by incubation in 100 mM NaCl, 10 mM Tris HCl pH 7.4, 1 mM EGTA, 2 mM MgCl$_2$, 1% Triton X-100, and Halt™ Protease Inhibitor Cocktail (Thermo Scientific) for 30 min at 4 °C. The total cellular cholesterol concentration was measured using the Amplex® Red Cholesterol Assay Kit (Thermo Scientific). The values were normalized to total cellular protein concentration, which was determined by BCA assay (Thermo Scientific).

The cholesterol content of the plasma membrane was measured using a protocol modified from Chu et al.[39]. In brief, cells were extensively washed with ice-cold assay buffer (310 mM sucrose, 1 mM MgSO$_4$, 0.5 mM Sodium phosphate [pH 7.4]) and then incubated with or without 1 U ml$^{-1}$ cholesterol oxidase for 3 min at room temperature. The buffer was removed and the cells washed once with ice-cold assay buffer. Cells were lysed and the cholesterol concentration measured as described above. The plasma membrane cholesterol concentration was calculated by subtracting the amount of intracellular cholesterol (cells incubated with cholesterol oxidase) from the total amount of cholesterol (cells incubated in the absence of cholesterol oxidase). The values were normalized to total cellular protein concentration determined by BCA assay.

**Cholesterol trafficking**. Unlabeled LDL (1 mg) from human plasma (Thermo Scientific) was incubated with 50 nmol cholesterol (Top-Fluor cholesterol, Avanti Polar Lipids) for 2 h at 40 °C and dialyzed overnight in PBS supplemented with 1 mM EDTA. LDL-deprived serum was prepared as described previously[54]. Cells were cultured in medium prepared with LDL-deprived serum for 24 h. Cholesterol trafficking was monitored by adding LDL complexed with Top-Fluor Cholesterol to the cells followed by incubation for 2 h. Cells were washed with culture medium and fixed with 4% PFA after various times of incubation in LDL-free medium.

**Cholesterol loading of plasma membrane**. Methyl-β-cyclodextrin (150 mg, MBCD, Sigma) was mixed with 5 mg cholesterol (Sigma) in 1 ml PBS and sonicated for 5 min (45% duty cycle, Branson Sonifier 250). Cells were incubated for 1 h at 37 °C with 1.5 mg ml$^{-1}$ MBCD and 50 μg ml$^{-1}$ cholesterol in serum-free DMEM medium.

**Western blot analysis**. Downregulation of clathrin heavy chain or STIM1 was evaluated by lysing cells in 100 mM NaCl, 10 mM Tris HCl pH 7.4, 1 mM EGTA, 2 mM MgCl$_2$, 1% Triton X-100, and Halt™ Protease Inhibitor Cocktail (Thermo Scientific) for 15 min at 4 °C. Lysates were cleared by a 15-min centrifugation at 16,000 × g at 4 °C. The subcellular localization of TFEB was evaluated by preparing the cells as described previously[55]. Protein concentration was determined with the BCA assay kit. Western blots were performed as described previously[56]. Signals were visualized with a chemiluminescence substrate (SuperSignal West Dura) or acquired with an Odyssey ClX (Li-COR) instrument. Signal intensities were quantified using ImageJ software. Uncroped western blots are presented in Supplementary Fig. 5.

**Total internal reflection fluorescence microscopy**. TIRF experiments were performed on fibroblasts transfected with vectors expressing STIM1-mCherry, using a previously described protocol[57]. Analyses were performed using ImageJ

software. The TIRF signal was obtained by thresholding and the area containing the TIRF signal normalized to the surface for each cell.

**Statistics and data analysis**. All statistical tests were performed using GraphPad Prism 6 and the tests are described in the figure legends. Multiple comparisons were performed using ANOVA when data had a normal distribution. Holm–Sidak multiple comparison tests allowed to compare the means of the different sets of data. $P < 0.05$ was considered to be statistically significant.

**Ethical approval**. The care and treatment of animals followed European legislation (N° 2010/63/UE) and national (Ministère de l'Agriculture, France) guidelines for the detention, use, and ethical treatment of laboratory animals. All experiments on animals were approved by the local ethics committee (approval APAFIS-5199) and conducted by authorized personnel.

**Reporting summary**. Further information on research design is available in the Nature Research Reporting Summary linked to this article.

## Data availability

The data that support the findings of this study are available in Supplementary Data 1.

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

## Acknowledgements

We thank Phenoparc, IGenSeq, Celis, and the ICM.quant facilities of the Institut du Cerveau et de la Moelle Épinière for their contributions. The 1D4B monoclonal antibody was obtained from the Developmental Studies Hybridoma Bank (University of Iowa, Department of Biology, IA 52242). This work was supported by the "Investissements d'avenir" program grants [ANR-10-IAIHU-06] and [ANR-11-INBS-0011] and received funding from the Verum Foundation (to A.B. and G.S.), the French Agency for Research (ANR) (to G.S.), the GIS-Maladies Rares Foundation (to G.S.), the Fondation Roger de Spoelberch (to A.B.), and the European Union with the ANR (to A.B., Seventh Frame-work Programme - FP7, Omics call; to G.S., the E-Rare programme) and the European Research Council (European Research Council Starting [grant No 311149] to F.D.). M.B. received a fellowship from the French Ministry of Research (Doctoral School ED3C). A. P. received an ARDoC fellowship (17012953) from the Région Ile de France (Doctoral School ED3C).

## Author contributions

M.B., A.P., A.B., G.S., and F.D. conceived and designed the experiments. M.B., A.P., R.M., J.B., M.H., Y.I., E.B., K.H.E.H., and F.D. performed the experiments. M.B., A.P., R.M., and F.D. analyzed the data. M.B. and F.D. wrote the paper with comments of all co-authors.

## Competing interests

The authors declare no competing interests.
