## [Peer Review File · Communications Biology]

Reviewers' comments:

Reviewer #1 (Remarks to the Author):

Nice paper. Please see the attached word file for my report. I am ok with my identity being known.

Reviewer #2 (Remarks to the Author):

In this study, Boutry et al. report an impairment in cholesterol trafficking in response to loss of the SPG11 protein spatacsin. They show that loss of spatacsin inhibits formation of tubules on late endosomes/lysosomes and prevents clearance of cholesterol from these compartments. This leads to lower cholesterol levels in plasma membrane and enhanced entry of calcium via SOCE, resulting in higher cytosolic calcium levels and the inhibition of TFEB nuclear translocation. The formation of tubules and lysosomal accumulation of cholesterol can be normalized via lowering intracellular calcium levels or downregulation of TFEB.

Overall, this work appears well done, and I have only a question (not a required stipulation) to address:

1. Did the authors investigate any cells from SPG15 patients for any of the studies shown? If so, this could be a valuable confirmation.

Reviewer #3 (Remarks to the Author):

The present article brings new understanding of the intracellular function of spatacsin, the protein deficient in spastic paraplegia 11 (SPG11), and of the consequences of its dysfunction in Spg11 ^{-/-} fibroblasts and neurons. Through a combination of experimental approaches, the authors demonstrate the involvement of spatacsin in the egress of cholesterol from LAMP-1 positive organelles, and provide several pieces of data supporting that this process depends on LE/lysosomal tubulation. In Spg11 ^{-/-} fibroblasts, failure to export cholesterol from LE/lysosomes leads to lower plasma membrane cholesterol, which activates store-operated calcium entry in the cytosol. The authors associate this finding to a higher level of TFEB in the nuclei of Spg11 ^{-/-} cells. Moreover, they demonstrate that reducing intracellular Ca⁺⁺ levels in Spg11 ^{-/-} cells or depleting TFEB by siRNA partially correct the LE/lysosomal tubulation defect and restores normal cholesterol levels in LE/lysosomes. These findings link calcium homeostasis and TFEB to the accumulation of cholesterol observed in Spg11 ^{-/-} fibroblasts. A few experiments support that deregulation of calcium homeostasis secondary to cholesterol accumulation also occurs in neuronal cells.

Taken together, the presented data provide novel and interesting information on the pathological alterations that may lead to neuron dysfunction in spastic paraplegia 11, and highlight ways to clear an excess of cholesterol from lysosomes in Spatacsin deficient cells (in vitro). I am convinced that this well-written article will be of great interest to scientists working on lysosomes and on lysosome-associated pathologies. There are only a few points that I recommend addressing prior to publication:

- an analysis of the activity level of mTOR and of calcineurin would strengthen the conclusions regarding TFEB activation and nuclear translocation in Spg11^{-/-} cells.

- The authors show that downregulation of TFEB promotes clearance of cholesterol from lysosomes in

Spg11 $-/-$ cells. This observation is intriguing considering that recent publications highlight that TFEB activation promotes cellular clearance in several lysosomal storage disease models. This is worth pointing out as it might influence thinking in the field.

Minor points:

- The first sentence of the result section is very vague. I recommend to either precise its meaning or to remove the sentence.

- Figure 1 C and D. According to the graph, the filipin staining intensity is similar in control and Spg11 $-/-$ cells. However, the signal in panel C appears weaker in Spg11 $-/-$ compared to control cells. If this cell is not representative of the whole cell population, the image should be replaced.

- p.8. It is stated that "Under these conditions, SOCE decreased markedly in both Spg11 $+/+$ and Spg11 $-/-$ fibroblasts (data not shown), along with resting cytosolic calcium level (Fig. 5A)." There is no statistical analysis in Fig 5A supporting the decrease of resting calcium level in Spg11 $+/+$ cells. The statement should be qualified.

Reviewer 1:

1) The loss of spatacsin (SPG11^{-/-}) is associated with cholesterol accumulation in late endosomes. Authors report ~ 4% increase in filipin (free cholesterol marker) and Lamp1 (lysosomal/endosomal marker) colocalization in SPG11^{-/-} vs. SPG11^{+/+} mouse fibroblasts. A similar increase (using GFPD4 rather than filipin) was also found in SPG11^{-/-} mouse primary cortical neurons vs. controls. Interpretation of the later results, however, is unclear because D4 is used to visualize PM cholesterol (and not endosome/lysosome cholesterol). Note: in figure legend 3A the authors highlight that GFP-D4 marks PM Cholesterol. Was D4H used? D4H stains cholesterol in endosomes (Ref 1).

- Do SPG11^{-/-} primary neurons also show increased filipin and LAMP1 colocalization?
- Previously the authors reported that glycosphingolipid accumulation also occurs in the absence of spatacsin (REF 2) and it is known that in some lysosomal storage diseases that the accumulation of glycol(sphingolipids) often results in the aberrant accumulation of additional lipid species such as cholesterol (for review please see Ref 3). Have the authors examined cholesterol accumulation in SPG11^{-/-} cells after inhibition of glycosphingolipid synthesis (i.e., using glucosylceramide synthase inhibitor in REF 2) to determine if the cholesterol accumulation is secondary to the glycosphingolipid storage? If this has not been investigated, then possibility of cholesterol storage being secondary to glycosphingolipid accumulation should be mentioned in the discussion (especially for readers not familiar with lysosomal storage disorders).

Response: The GFP-D4 probe can be used to label plasma membrane cholesterol on live cells. However, on fixed and permeabilized cells, it can also label the intracellular cholesterol (including in lysosomes), similar to filipin. In neurons, we monitored the accumulation of cholesterol with the GFP-D4 probe. We have now also monitored this accumulation using filipin, and we obtained similar results to the ones obtained with GFP-D4. As illustrated in Supplementary Fig. 1d, we observed higher amount of colocalization of filipin staining with LAMP1 in Spg11 knockout neurons compared to control neurons. Furthermore, when we inhibited glycosphingolipid synthesis with miglustat, we did not prevent the cholesterol accumulation in lysosomes in Spg11 knockout neurons (Supplementary Fig. 1d). We have modified the text (p. 5) accordingly.

2) Inhibition of tubule formation in endosomes/lysosomes leads to cholesterol accumulation. Authors report that knockdown of clathrin heavy chain (CHC) in SPG11^{+/+} mouse fibroblasts decreases the number of LAMP1 positive tubules. Similar to major claim #1, CHC KD leads to ~ 3% increase in filipin and LAMP1 colocalization. Inhibiting dynamin, the binding partner of spatacsin, with dynasore also increased filipin/LAMP1 colocalization in SPG11^{+/+} mouse fibroblasts. Interestingly, dynasore did not have a similar effect in SPG11^{-/-} mouse fibroblasts; thus, suggesting that spatacsin and dynamin regulated the same intracellular pool of cholesterol.

3) Cholesterol levels are lower in the PM of SPG11^{-/-} mouse fibroblasts.

These findings corroborate an earlier finding that implicates spatacsin as an important regulator of cholesterol transport from the lysosomes to the PM (manuscript Ref #35). Authors report that GFP-D4 staining, a marker of PM cholesterol, is reduced in SPG11^{-/-} vs. SPG11^{+/+} mouse fibroblasts. It is also shown that knockdown of CHC leads to similar reduction in GFP-D4 staining.

- *It is not stated in the text or figure legend 3 if the latter experiment (CHC KD leads to reduced GFP-D4 staining) was carried out in SPG11^{-/-} or SPG11^{+/+} fibroblasts. Please clarify.*

Response: We have clarified in legend of Figure 3 (p24) that experiments with knockdown of CHC or dynasore treatment were performed on control (*Spg11*^{+/+}) mouse fibroblasts.

4) Reduced PM cholesterol alters store-operated calcium entry (SOCE).

*Using electron microscopy, the authors show that there is an increase in membrane contacts between the ER and PM in the cortex of SPG11^{-/-} mice. These contacts are known to play a role in regulating the transfer of lipids and Ca⁺⁺ between the ER and PM (manuscript refs 24, 25). Additional experiments showed that STIM1-mCherry fluorescence was elevated in SPG11^{-/-} mouse fibroblasts vs. controls. STIM1 senses reductions in ER Ca⁺⁺ and interacts with the PM Ca⁺⁺ channel *Orai1* to import extracellular Ca⁺⁺. The latter results suggest that there are more ER/PM contact sites in SPG11^{-/-} mouse fibroblasts. These findings corroborate their EM findings in mice. Additional experiments showed that depletion of intracellular Ca⁺⁺ with thapsigargin led to increased detection of Fura-2, a marker of intracellular Ca⁺⁺ after the addition of CaCl₂ to the cell medium. Finally, the authors showed that increasing PM cholesterol with MBCD normalized SOCE in SPG11^{-/-} mouse fibroblasts to SPG11^{+/+} levels.*

- *These findings indicate that SOCE is altered in SPG11^{-/-} mouse fibroblasts, which is novel. However, a number of studies (e.g., Refs 4-9) have shown that PM cholesterol impacts SOCE in various cellular systems. Some studies suggest that cholesterol enhances SOCE whereas other indicates that it inhibits SOCE. It would be helpful if the authors elaborated more on their findings in relation to previous studies.*

Response: Indeed cholesterol depletion led to various consequences on SOCE in different cellular systems. We have thus expanded the discussion on this point (p11-12).

5) High SOCE in the absence of spatacsin increases Ca⁺⁺ cytoplasmic levels. The authors showed that cytosolic Ca⁺⁺ levels were significantly higher in SPG11^{-/-} vs. SPG11^{+/+} mouse fibroblasts. Additional experiments demonstrated that knockdown of STIM1 reduced SOCE equivalently in SPG11^{-/-} and SPG11^{+/+} mouse fibroblasts and that increasing PM cholesterol levels with MBCD normalized intracellular Ca⁺⁺ levels in SPG11^{-/-} fibroblasts to levels in SPG11^{+/+} fibroblasts. These findings nicely support that notion that reduced PM cholesterol in SPG11^{-/-} mouse fibroblast increases SOCE (major finding #4), which leads to an aberrant elevation in intracellular Ca⁺⁺.

6) High cytosolic Ca⁺⁺ leads to leads to increased nuclear translocation of TFEB in the absence of spatacsin. *It is well known that increasing intracellular Ca⁺⁺ induces the nuclear translocation of TFEB (Refs 10,11). The authors show that nuclear TFEB levels are significantly elevated in SPG11^{-/-} mouse fibroblasts compared to controls (which make sense given major claim #5). It is also shown that reducing Ca⁺⁺ with EGTA-AM, an intracellular Ca⁺⁺ chelator, normalized nuclear TFEB levels to levels in SPG11^{+/+} fibroblasts. Treatment with EGTA-AM also significantly increased the number of LAMP1 positive tubules in SPG11^{-/-} mouse fibroblasts vs. control SPG11^{-/-} mouse fibroblasts. Additional experiments showed knockdown of TFEB reduced nuclear TFEB levels in both SPG11^{-/-} and SPG11^{+/+} mouse fibroblasts. In SPG11^{-/-} mouse fibroblasts knockdown of TFEB increased the number of*

LAMP1 positive tubules, and reduced the number of tubules that colocalized with filipin. Moreover, the authors also show that knockdown of STIM1 in SPG11^{-/-} mouse fibroblasts reduces LAMP1/filipin colocalization and normalizes PM cholesterol (GFP-D4 staining) to levels in SPG11^{+/+} mouse fibroblasts.

- These findings appear to be at odds with TFEB's known role in promoting autophagy, lysosomal biogenesis and the metabolism of lipids. Can the authors clarify/discuss how reducing TFEB either directly or through modulation of intracellular Ca⁺⁺ via knockdown of STIM1 leads to an increase in LAMP1 given that TFEB stimulates autophagosome and lysosomal biogenesis? Can the authors also clarify how reducing TFEB nuclear translocation leads to a reduction in LAMP1/filipin colocalization (i.e., which presumably reflects increased lipid degradation) and increased PM cholesterol levels when it is known that TFEB stimulates lipid metabolism (REF 12)? Impairing TFEB should lead to reduced lipid metabolism.

- Have the authors considered the possibility that increased nuclear translocation of TFEB is a compensatory response in SPG11^{-/-} cells to restore PM cholesterol levels to their preferred homeostatic level? For example, TFEB is activated to stimulate the lysosomal hydrolysis/recycling of cholesterol enriched membranes, which generates a source of recycled cholesterol that can be returned to the PM. However, since spatacsin is missing this recycled cholesterol cannot be returned to the PM, which leads to it accumulating in endosomes/lysosomes.

Response: We have discussed the points highlighted by the reviewer (p12). It has been shown that TFEB enhanced lipid metabolism in liver. In the case of sterol metabolism, most of the enzymes that are modulated by TFEB are localized in the endoplasmic reticulum. It is therefore unlikely that TFEB directly regulates cholesterol degradation by lysosomes. In lysosomes, cholesterol is transferred by the proteins NPC1 and NPC2 from the lumen to the lysosome membrane, allowing its transfer to other subcellular compartment (Ikonen, 2018). The increase in TFEB nuclear translocation observed in *Spg11*^{-/-} cells could thus modulate the trafficking of cholesterol to allow its clearance from lysosomes. One possibility would be that nuclear translocation of TFEB could be a compensatory mechanism in *Spg11*^{-/-} cells. However, we observed that downregulation of TFEB promotes clearance of cholesterol from lysosomes, likely by restoring the tubulation of lysosomes which appears to be involved in cholesterol trafficking from lysosomes to other compartments. This suggests that nuclear translocation of TFEB inhibited cholesterol clearance in our model (see comments of Reviewer 3).

- In addition to promoting lipid metabolism, TFEB also suppresses de novo sterol synthesis (see REF 12). Did the authors investigate if de novo sterol synthesis was elevated in their experiments in which TFEB and STIM1 were knocked down? Perhaps increased de novo sterol synthesis (and nonvesicular sterol transport) led to the increased cholesterol levels in the PM of SPG11^{-/-} cells after TFEB knockdown.

Response: TFEB was indeed shown to promote lipid metabolism in liver. We have therefore checked the levels of active SREBP that is a transcription factor that regulates the expression of enzymes required for cholesterol synthesis. We observed no change in the levels of the active form of SREBP in control or *Spg11* knockout fibroblasts (Supplementary Fig. 4F). Furthermore, these levels were not significantly modified when TFEB was downregulated, suggesting that this pathway is not directly affected by

TFEB. We can however not exclude that others pathways regulating cholesterol synthesis may be affected by TFEB or the loss of spatacsin.

Are the claims convincing? If not, what further evidence is needed?

• *It is clear from experiments carried out to support claim #6 that reductions in PM cholesterol levels enhances SOCE to increase intracellular Ca^{++} to increase the nuclear translocation of TFEB in SPG11^{-/-} mouse fibroblasts. Did the authors investigate if calcineurin (REF 10), which is activated is by Ca^{++} and dephosphorylates TFEB to allow it to enter the nucleus is also altered in SPG11^{-/-} mouse fibroblasts? Was TFEB nuclear translocation investigated following the direct manipulation of calcineurin? Evidence of altered calcineurin signaling would strengthen the authors findings that increased TFEB nuclear translocation is Ca^{++} dependent.*

Response: We have monitored the role of calcineurin in the regulation of TFEB nuclear translocation, using siRNA to downregulate its expression. Downregulation of calcineurin reduced the amount of nuclear TFEB in Spg11^{-/-} fibroblasts (Fig. 6), suggesting that the action of calcium to regulate the translocation of TFEB is dependent on calcineurin.

• *It is possible that altered intracellular cholesterol homeostasis may also influence Ca^{++} independent known regulators of TFEB nuclear translocation (i.e., mTORC1, ERK2/MAPK1, GSK3B and AKT, see Ref #13). Showing that these regulators are not altered in SPG11^{-/-} would strengthen the supposition that Ca^{++} dependent pathways are responsible for disease related changes in TFEB signaling. Alternatively, the authors need to make readers (especially neuromuscular disease audience) aware of these alternative activation pathways, their potential involvement, and if they should be investigated in the future.*

Response: We agree with the reviewer that calcium-independent regulators of TFEB could play a role in our observations. We have mentioned in the discussion the various cellular pathways that could allow the phosphorylation of TFEB. The GSK3 pathway could be of particular interest, as it has been shown to be altered in iPS cells derived from SPG11 patients (Mishra et al, 2016). We have discussed this point in our revised manuscript. Since our data suggest that the nuclear translocation of TFEB is dependent on calcineurin in Spg11^{-/-} cells, we did not include data regarding the ERK, GSK3 and Akt pathways, as it may impair the clarity of the manuscript. We agree that it would be an important point to test in the future, and it would require a full characterization of the signaling pathways.

Reviewer 2:

In this study, Boutry et al. report an impairment in cholesterol trafficking in response to loss of the SPG11 protein spatacsin. They show that loss of spatacsin inhibits formation of tubules on late endosomes/lysosomes and prevents clearance of cholesterol from these compartments. This leads to lower cholesterol levels in plasma membrane and enhanced entry of calcium via SOCE, resulting in higher cytosolic calcium levels and the inhibition of TFEB nuclear translocation.

The formation of tubules and lysosomal accumulation of cholesterol can be normalized via lowering intracellular calcium levels or downregulation of TFEB.

Overall, this work appears well done, and I have only a question (not a required stipulation) to address:

1. Did the authors investigate any cells from SPG15 patients for any of the studies shown? If so, this could be a valuable confirmation.

Response: We agree with the reviewer that validating the results in SPG15 cells would be really interesting. Since all our experiments were performed in cells derived from a mouse model of Spg11, we would have preferred to use mouse embryonic fibroblasts derived from a Spg15 knockout mouse. However, we did not have access to these cells and could not perform the experiment. Nonetheless, we have added a sentence in the discussion to mention this point.

Reviewer #3 (Remarks to the Author):

The present article brings new understanding of the intracellular function of spatacsin, the protein deficient in spastic paraplegia 11 (SPG11), and of the consequences of its dysfunction in Spg11 -/- fibroblasts and neurons. Through a combination of experimental approaches, the authors demonstrate the involvement of spatacsin in the egress of cholesterol from LAMP-1 positive organelles, and provide several pieces of data supporting that this process depends on LE/lysosomal tubulation. In Spg11 -/- fibroblasts, failure to export cholesterol from LE/lysosomes leads to lower plasma membrane cholesterol, which activates store-operated calcium entry in the cytosol. The authors associate this finding to a higher level of TFEB in the nuclei of Spg11 -/- cells. Moreover, they demonstrate that reducing intracellular Ca⁺⁺ levels in Spg11 -/- cells or depleting TFEB by siRNA partially correct the LE/lysosomal tubulation defect and restores normal cholesterol levels in LE/lysosomes. These findings link calcium homeostasis and TFEB to the accumulation of cholesterol observed in Spg11 -/- fibroblasts. A few experiments support that deregulation of calcium homeostasis secondary to cholesterol accumulation also occurs in neuronal cells.

Taken together, the presented data provide novel and interesting information on the pathological alterations that may lead to neuron dysfunction in spastic paraplegia 11, and highlight ways to clear an excess of cholesterol from lysosomes in Spatacsin deficient cells (in vitro). I am convinced that this well-written article will be of great interest to scientists working on lysosomes and on lysosome-associated pathologies. There are only a few points that I recommend addressing prior to publication:

- an analysis of the activity level of mTOR and of calcineurin would strengthen the conclusions regarding TFEB activation and nuclear translocation in Spg11-/- cells.

Response: We have monitored the activity of mTOR using the phosphorylation state of the mTOR substrates S6 and S6 kinase. We did not detect any significant change in the levels of mTOR activity between control and Spg11 knockout fibroblasts (Supplementary Fig. 4a). As calcineurin is a calcium-dependent mediator of the nuclear translocation of TFEB, we evaluated whether nuclear translocation of TFEB is dependent on calcineurin in absence of spatacsin. Downregulation of calcineurin using siRNA reduced the amount of nuclear TFEB in Spg11-/- fibroblasts (Fig. 6), suggesting that the action of calcium to regulate the translocation of TFEB is dependent on

calcineurin. These data have been added in the result section (p. 9), and discussed (p. 12)

- The authors show that downregulation of TFEB promotes clearance of cholesterol from lysosomes in Spg11 -/- cells. This observation is intriguing considering that recent publications highlight that TFEB activation promotes cellular clearance in several lysosomal storage disease models. This is worth pointing out as it might influence thinking in the field.

Response: We thank the reviewer for this suggestion. We have indeed insisted on this point in the discussion (p. 12).

Minor points:

- The first sentence of the result section is very vague. I recommend to either precise its meaning or to remove the sentence.

Response: We have actually removed the first sentence of the result section.

- Figure 1 C and D. According to the graph, the filipin staining intensity is similar in control and Spg11 -/- cells. However, the signal in panel C appears weaker in Spg11 -/- compared to control cells. If this cell is not representative of the whole cell population, the image should be replaced.

Response: We have replaced images in Fig. 1c.

- p.8. It is stated that “Under these conditions, SOCE decreased markedly in both Spg11 +/+ and Spg11 -/- fibroblasts (data not shown), along with resting cytosolic calcium level (Fig. 5A).” There is no statistical analysis in Fig 5A supporting the decrease of resting calcium level in Spg11 +/+ cells. The statement should be qualified.

Response: We have added in Fig. 5a the statistical analysis showing that decreasing extracellular calcium levels decreased resting calcium level in Spg11 +/+ cells.

REVIEWERS' COMMENTS:

Reviewer #1 (Remarks to the Author):

I accept the changes made by the authors. I support the publication of this manuscript.

James Dodge

Reviewer #3 (Remarks to the Author):

The authors have addressed all of my concerns. I recommend publication of the article.